# A systematic review of antibody mediated immunity to coronaviruses: kinetics, correlates of protection, and association with severity

Angkana T. Huang [1,2,3,11], Bernardo Garcia-Carreras[1,2,11], Matt D. T. Hitchings[1,2,11], Bingyi Yang [1,2,11], Leah C. Katzelnick [1,2,11], Susan M. Rattigan[1,2], Brooke A. Borgert [1,2], Carlos A. Moreno[1,2], Benjamin D. Solomon[4], Luke Trimmer-Smith[1,2], Veronique Etienne[2,5], Isabel Rodriguez-Barraquer[6], Justin Lessler [7], Henrik Salje[1,8,9], Donald S. Burke [10], Amy Wesolowski [7] & Derek A. T. Cummings [1,2✉]

Many public health responses and modeled scenarios for COVID-19 outbreaks caused by SARS-CoV-2 assume that infection results in an immune response that protects individuals from future infections or illness for some amount of time. The presence or absence of protective immunity due to infection or vaccination (when available) will affect future transmission and illness severity. Here, we review the scientific literature on antibody immunity to coronaviruses, including SARS-CoV-2 as well as the related SARS-CoV, MERS-CoV and endemic human coronaviruses (HCoVs). We reviewed 2,452 abstracts and identified 491 manuscripts relevant to 5 areas of focus: 1) antibody kinetics, 2) correlates of protection, 3) immunopathogenesis, 4) antigenic diversity and cross-reactivity, and 5) population seroprevalence. While further studies of SARS-CoV-2 are necessary to determine immune responses, evidence from other coronaviruses can provide clues and guide future research.

[1] Department of Biology, University of Florida, Gainesville, FL, USA. [2] Emerging Pathogens Institute, University of Florida, Gainesville, FL, USA. [3] Department of Virology, Armed Forces Research Institute of Medical Sciences, Bangkok, Thailand. [4] National Human Genome Research Institute, National Institutes of Health, Bethesda, MD, USA. [5] Department of Comparative, Diagnostic & Population Medicine, University of Florida, Gainesville, FL, USA. [6] Division of HIV, ID, and Global Medicine, University of California, San Francisco, CA, USA. [7] Department of Epidemiology, Johns Hopkins Bloomberg School of Public Health, Baltimore, MD, USA. [8] Department of Genetics, University of Cambridge, Cambridge, UK. [9] Mathematical Modelling of Infectious Diseases Unit, Institut Pasteur, Paris, France. [10] Department of Epidemiology, University of Pittsburgh, Pittsburgh, PA, USA. [11] These authors contributed equally: Angkana T. Huang, Bernardo Garcia-Carreras, Matt D. T. Hitchings, Bingyi Yang, Leah C. Katzelnick. ✉email: datc@ufl.edu

A pandemic of severe acute respiratory syndrome coronavirus 2 (SARS-CoV-2[1–3]) is currently underway, and is resulting in severe morbidity and mortality worldwide. Limited pre-existing immunity to this virus is thought to be responsible for the explosive increase in cases. Nearly all transmission models of SARS-CoV-2 assume that infection produces immunity to reinfection for durations of at least 1 year[1–3]. This assumption is relevant to public health officials implementing and managing various nonpharmaceutical interventions, the utility of sera from infected individuals as a therapeutic[4], and the ability for serological tests to identify those who are immune[5]. The dynamics of immunity will also affect the performance of serological testing to quantify the extent of infection in populations. However, knowledge of the dynamics and nature of immune response to SARS-CoV-2 infection is limited, and the scientific basis for durable immunity, upon which these key public health and clinical strategies are dependent, is not well developed.

In the context of our current limited understanding of SARS-CoV-2 immunity, this review looks for insights from studies of the broader coronavirus family. Several authors have noted human experimental infection studies (called human challenge studies), suggesting that protection after coronavirus infections may last for only 1 or 2 years[6–9]. Human coronaviruses (HCoVs) have been used in human challenge experiments since shortly after their discovery in 1965[10,11]. These experiments, where individuals were intentionally infected with HCoV, provide some of the clearest characterization of human responses to coronaviruses and the potential for immune responses to limit infection and disease. Multiple human challenge studies measured antibody immunity before a coronavirus challenge and identified antibody responses that were associated with protection from infection, serological response, or symptoms[12]. The low severity of HCoV allowed for safe use of these viruses[6–9] as human challenge. The greater likelihood of severe illness in SARS-CoV-2 limits the applicability of such experiments, although some have argued for their use in subsets of the population[13].

The duration of immunity to SARS-CoV-2 will dictate the overall course of the pandemic and post-pandemic dynamics[7], and so an understanding of the temporal dynamics of protective immunity is critical. As with other introductions of novel pathogens[14], explosive outbreaks of SARS-CoV-2 across the globe may threaten its persistence by reducing the number of available hosts susceptible to infection. Immune interactions with endemic coronaviruses could theoretically affect the short- and long-term dynamics of SARS-CoV-2, and vice versa[15,16] through cross-protection or antibody-dependent enhancement[17], but these interactions and effects are not yet understood. If SARS-CoV-2 does become endemic, age-stratified seroprevalence studies of endemic coronaviruses may provide insight into incidence rates in the presence of higher levels of population immunity.

Here, we describe the results of a systematic review of literature on antibody measures of immunity to coronaviruses, including endemic HCoV (principally HCoV-229E, HCoV-HKU1, HCoV-OC43, and HCoV-NL63), SARS-CoV (severe acute respiratory syndrome coronavirus that emerged in 2002), MERS-CoV (Middle East respiratory syndrome coronavirus), and early work on SARS-CoV-2. We conceptualize the stages of exposure and infection at which immunity may play a role in the dynamics of SARS-CoV-2, and how literature describing work on this and other coronaviruses can provide insights into these stages, as follows (see Fig. 1, a visual abstract of our review). First, an exposure to a pathogen generates an antibody response that changes over time and between individuals (antibody kinetics). Upon exposure, infection history might play a role in providing protection against new infection, and the literature can provide

evidence for such correlates of protection through challenge studies and longitudinal cohort studies. Upon infection, an individual's immune state, possibly impacted by pre-existing antibodies to other coronaviruses (among other mechanisms), may cause harm through immunopathogenesis, and the literature on this topic primarily consists of in vitro experiments. Correlates of immunity (or risk through immunopathogenesis) are complicated by the existence of multiple genera of coronaviruses that are antigenically diverse and may provide cross-protection, and which may also cause false-positive assay results due to cross-reactivity. Finally, the preceding phenomena at the individual level interact to determine population seroprevalence. Studies that measure these quantities across different age groups can provide evidence, albeit ecological evidence, for or against the proposed mechanisms of immunity at the individual level.

## Results

**Paper identification**. Our searches identified 2452 abstracts of potential relevance (Supplementary Fig. 1). Two reviewers read each abstract and identified 491 papers for full review. Papers were classified into our areas of focus. Below, we present the findings of our review for each area of focus.

**General background: serological assays**. Multiple serological assays have been used to characterize antibody responses to coronaviruses. Assays we encountered in our review fell into two major categories (Supplementary Table 1). The most commonly used assays were binding assays, including enzyme-linked immunosorbent assays (ELISA), immunofluorescence assays (IFA), western blots, and complement fixation (CF). Hemagglutination inhibition assays (HAI), which measure the ability of antibodies in sera to prevent binding of virus to red blood cells, have previously been used for coronaviruses, but are no longer common. The final category of assays was neutralization assays. Neutralization assays are typically considered the gold standard for measuring functional antibody responses as they measure biological activity throughout the viral replication process. Researchers have used assays of both types to characterize antibody activity by antibody class (i.e., IgG, IgM, and IgA). As these antibodies are known to have different temporal dynamics, we reported when specific classes of antibodies were characterized in the figures. If no characterization was specified, we reported the measures of antibody aggregated across all classes. The source of samples in the reviewed studies was almost exclusively serum samples. However, mucosal samples collected by swab or nasal washings were also reported.

**Antibody responses: kinetics and clinical severity**. Initially, 164 studies were classified as relevant to antibody kinetics and to the association of antibody responses with clinical severity. Of these, 58 were selected after further review. Supplementary Table 2 provides summaries for some of these studies. We digitized data from a subset of 51 studies that included sufficient detail on longitudinal antibody measurements (Supplementary Data 1; 5 on endemic HCoVs, 11 on MERS-CoV, 34 on SARS-CoV, and 2 on SARS-CoV-2 (one of which is non-peer-reviewed preprint)). In all, 8% of studies reported cumulative seroconversion only. In total, 60% of the digitized studies provided estimates during the first week after the onset of symptoms, while 75% had measurements at least 1 month after the onset of symptoms[18].

In the studies we shortlisted, antibody responses to infecting coronaviruses were rarely reported during the acute phase of illness (1–7 days)[19–26]. Corman et al.[26] detected antibodies in both ELISA and neutralization tests in 24 of 27 MERS-CoV patients within the first week. Many studies describe an immune

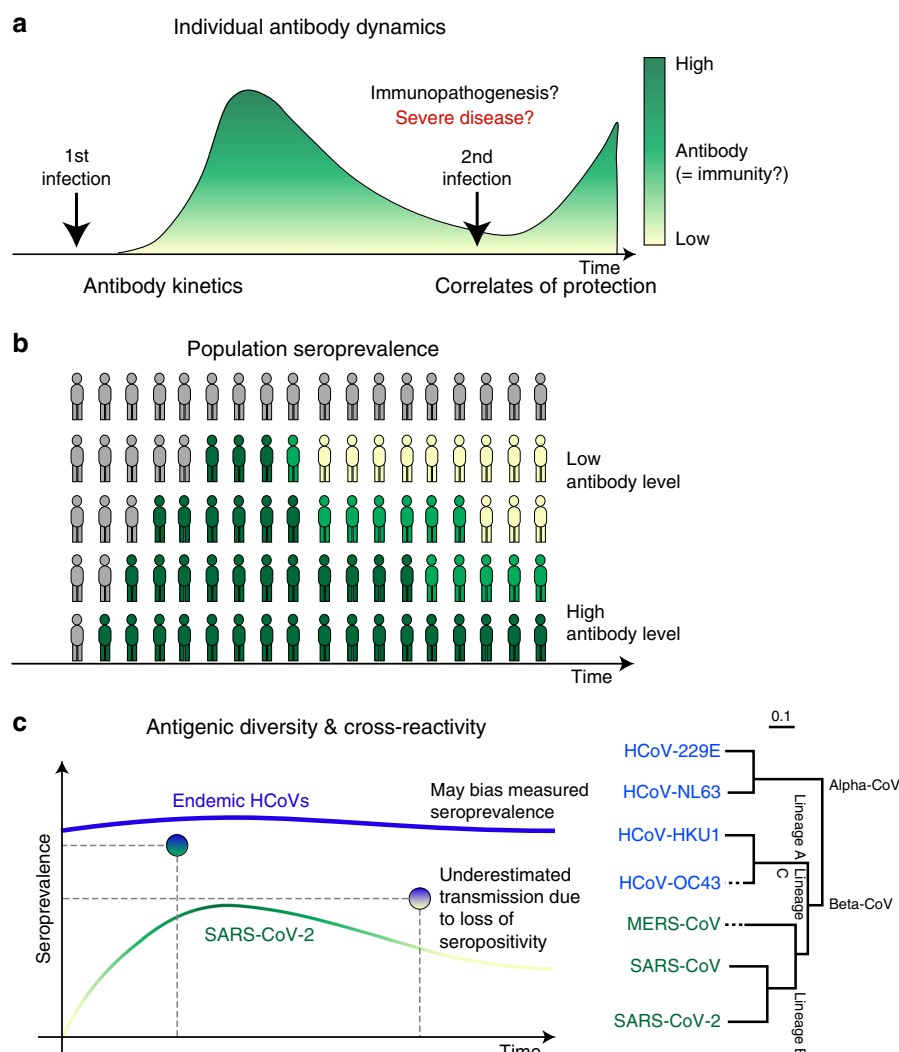

**Fig. 1 Aspects of antibody response included in this review.** This figure shows the areas of focus of our review within our conceptualization of the stages of exposure and infection at which we believe that antibody-mediated immunity may play a role in the dynamics of severe acute respiratory syndrome coronavirus 2 (SARS-CoV-2). At the individual level (**a**), antibody response following the first infection/exposure increases and then declines (antibody kinetics). Sometime later, individuals may be exposed to SARS-CoV-2 again. They may be protected from infection by their acquired immunity (correlates of protection). Their acquired immunity may also moderate the severity of infection with some possibility that pre-existing immunity may lead to immunopathogenesis (relevant to both first and second exposure). These individual-level dynamics aggregate to form the population-level seroprevalence. **b** Measures of seroprevalence may imperfectly measure past exposure to infection due to antigenic diversity of future SARS-CoV-2 viruses and cross-reactivity of endemic human coronaviruses (HCoVs) with SARS-CoV-2. **c** Measures of seroprevalence may also be inconsistent across times, as antibody levels within individuals wane.

response, characterized by a robust increment of antibody titers for HCoV-229E, MERS-CoV, SARS-CoV, and SARS-CoV-2 after the second or third week following the onset of illness[8,20,23,26–37].

Callow et al.[8] found similar dynamics in IgA and IgG across ten individuals experimentally infected with HCoV-229E: antibody levels increased after 8 days and peaked around 14 days, although significant variation between patients was reported. Yang et al.[21], analyzing the data across 67 patients, found higher positive rates of IgM against SARS-CoV than IgG during the first month. The proportion of patients who seroconverted for IgM peaked 30 days after onset, followed by a gradual decrease of IgM levels, while IgG levels peaked by week 25[21]. Using sera from 18 SARS-CoV patients, Mo et al.[23] noted that IgM, IgG, and neutralizing antibodies increased after day 15, and while IgM and neutralizing antibodies peaked on day 30, IgG peaked on day 60. Another study on 30 SARS-CoV-infected patients[20] detected seroconversion of IgM, IgG, and IgA at similar times, indicating

that the earliest they reached a peak was on average 15 days[38]. While it is currently too early to characterize how anti-SARS-CoV-2 antibodies will change over prolonged periods of time, preliminary studies have analyzed antibody changes in recent infections. Tan et al.[39] found that IgM was detected on day 7 and peaked on day 28 (across 28 patients), and IgG appeared by day 10 and peaked on day 49 (45 patients), while Zhao et al.[40] determined that seroconversion among 173 patients took place at median times of 12 (IgM), 14 (IgG), and 11 (neutralizing antibodies) days.

Multiple studies reported that while IgM and IgG titers increased during the first week following symptom onset, IgM levels gradually waned (while remaining detectable) for SARS-CoV and MERS-CoV in comparison to IgG levels about a month post follow-up[20,23,27]. Most studies that examined antibody kinetics over extended periods of time focused on IgG[20,23,41–48]. Callow et al.[8] found among experimental infections with

HCoV-229E that after peaks in IgG and IgA, antibody levels waned, and between 11 weeks and 1 year post inoculation, were at similar levels to those found in inoculated but uninfected patients. Other studies[41,46] reported detectable IgG levels in recovered MERS-CoV patients, respectively, at 5 months and 1 year after illness onset, while another[21] detected IgG antibodies across 67 SARS-CoV patients after 82 weeks, the endpoint of the study. We found few studies that analyzed changes in antibody kinetics over the course of many years after illness onset[42,43]. The 18 SARS-CoV patients in Mo et al.[23] were followed for 2 years; in that study, after peaking, IgM levels were undetectable by day 180. On the other hand, levels of IgG were still high on day 180 and gradually declined to still detectable levels by day 720, while neutralizing antibodies were detectable in 17 of 18 patients at day 720, but at low titers[23]. Cao et al.[43] described similar long-term dynamics for IgG and neutralizing antibodies over the course of a 3-year study on SARS-CoV: titers for both peaked at month 4, and while they waned thereafter, 74.2% and 83.9% of patients had detectable levels of IgG and neutralizing antibodies, respectively, at month 36. Liu et al.[47] found that a high proportion of 19 recovered SARS-CoV patients were positive up to 2 years post infection, with the percentage declining in the third year, a pattern similar to that found by Wu et al.[49] in 18 patients[50].

Some studies have explored a potential link between case severity and antibody response; however, the available information analyzing the nature of this relationship is uneven across viruses. Several studies on MERS-CoV as well as the first preliminary analyses on SARS-CoV-2 explicitly explore this connection, while fewer studies on endemic HCoVs and SARS-CoV do so. Several studies found that cases of varying severity (e.g., asymptomatic, mild, and severe) developed detectable antibodies against HCoV-229E, SARS-CoV, SARS-CoV-2, and MERS-CoV[28,30,39,43,46]. One study[51] found that the rise in antibodies between acute and convalescent sera correlated positively with symptoms and clinical score in 15 patients experimentally infected with HCoV-229E (Table 1). However, most studies of SARS-CoV did not report symptom severity, and evidence of differences in antibody responses among cases experiencing symptoms of different severity is inconclusive. SARS-CoV survivors with sequelae were found to have lower neutralizing antibodies than patients without sequelae[43], although the same study otherwise found no significant differences in kinetics according to disease severity. Chan et al.[45] found no evidence of difference in antibody responses between patients who survived or died[52,53]. In MERS-CoV, on the other hand, Ko et al.[28] found that both seroconversion rate and peak antibody levels increased with disease severity, while Okba et al.[54] reported a robust response to severe infections as opposed to low or no seroconversion in asymptomatic and mild cases. Milder infections also appeared to be less likely to elicit serologic responses[46,54], although Okba et al.[54] suggest that seroconversion detection may depend on the antibodies being specifically assayed. More severe cases were also found to have slower responses[28,30]; 75% of patients who died had not seroconverted by week 3[28]. Some authors have hypothesized[54,55] that seroconversion rates in severe cases may be associated with prolonged viral shedding, and that low antibody responses in mild cases may be due to short-lived infections. Another study[46] suggested that weaker antibody responses to endemic HCoVs (specifically HCoV-229E) might be because these mainly infect the upper respiratory tract. Preliminary studies on SARS-CoV-2 point to a possibly contrasting pattern to MERS-CoV: while IgM antibodies appear at the same time in severe and nonsevere cases, IgG appears sooner in severe cases[39]. On the other hand, neutralizing antibody titers were higher in severe cases[40].

The distributions of time points at which antibodies were detected (see "Methods") in the digitized data are shown in Fig. 2. The median time to detection across different antibodies was the shortest for SARS-CoV-2 (11.0 days; interquartile range (IQR) 7.0–14.0 days), followed by SARS-CoV (14.0 days; IQR 10.0–18.0 days) and MERS-CoV (15.0 days; IQR 12.0–18.0 days). Severity appears to be associated with time to detection of IgM in MERS-CoV cases only (2 days longer), and IgG in both MERS-CoV and SARS-CoV-2 (2–3 days longer for more severe cases). All data on time to seroconversion were based on symptomatic patients. No data were available for asymptomatic individuals.

Figure 3 provides a broad sense for trajectories of antibodies (also see Supplementary Figs. 3 and 4). Most longer-term studies (>10 weeks) were for MERS-CoV and reported IgG and neutralizing antibodies; these showed the presence of IgG and neutralizing antibodies up to 60 weeks after symptom onset (Fig. 3). The studies reporting symptom severity with longer-term data focused on MERS-CoV. Not all of these studies reported a cutoff for the assay used; in those that did, two-thirds of patients with mild symptoms had detectable or positive IgG antibodies at 6 months and 1 year, while all patients with severe symptoms had detectable IgG antibodies at the same points in time (Table 2).

**Correlates of protection.** Identification of a correlate of protection requires characterization of immune responses prior to a known exposure or period of risk in which infection or illness outcomes are characterized. In our review, we found that this level of detail was only present in human challenge experiments with HCoVs. We identified 18 studies in which volunteers were exposed to experimental infections with HCoV. Of these, six associated pre-infection antibody measurements with virologic, serologic, or illness outcomes upon experimental infection (Supplementary Table 4).

The earliest identified experimental infection study of coronaviruses found that seven of eight subjects with neutralizing titer <5 excreted virus after experimental exposure compared to only one of four subjects with pre-exposure titer of 40 or greater[10].

---

**Table 1 Key questions for SARS-CoV-2.**

Key questions for SARS-CoV-2
- What are the kinetics of antibody-mediated immune responses to infection?
- Do people who have more severe disease mount stronger antibody responses after infection?
- How do antibody responses vary between different types of antibodies or as measured by different assays?
- How does the presence of antibodies impact the clinical course and severity of the disease?
- Is there cross-reactivity with different coronaviruses?
- Does cross-reactivity lead to cross-protection?
- Will infection protect you from future infection?
- How long will antibody-mediated immunity last?
- What are the antibody-based correlates of protection?

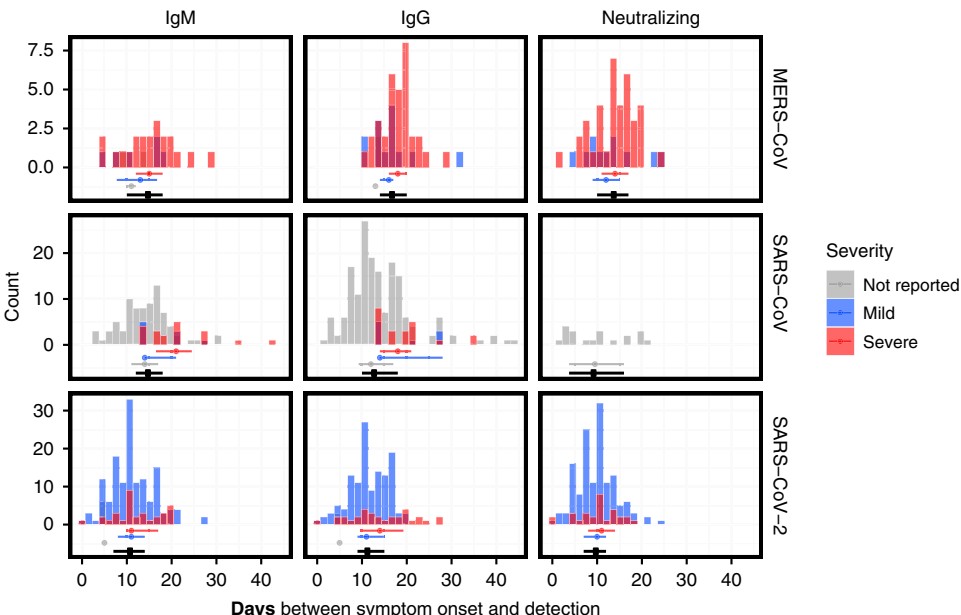

**Fig. 2 Distributions of times from symptom onset to detection of antibodies.** Times between symptom onset and the detection of IgM (left column), IgG (middle column), and neutralizing antibodies (right column), for MERS-CoV (upper row), SARS-CoV (middle row), and severe acute respiratory syndrome coronavirus 2 (SARS-CoV-2) (bottom row). Dots and lines below each histogram indicate the median values and interquartile range (IQR) across all severity ratings (black), mild symptoms (blue), severe symptoms (red), and no reported severity (gray). Data were digitized from 17 studies[19,20,24,25,27–30,32,33,38,40,46,147–150].

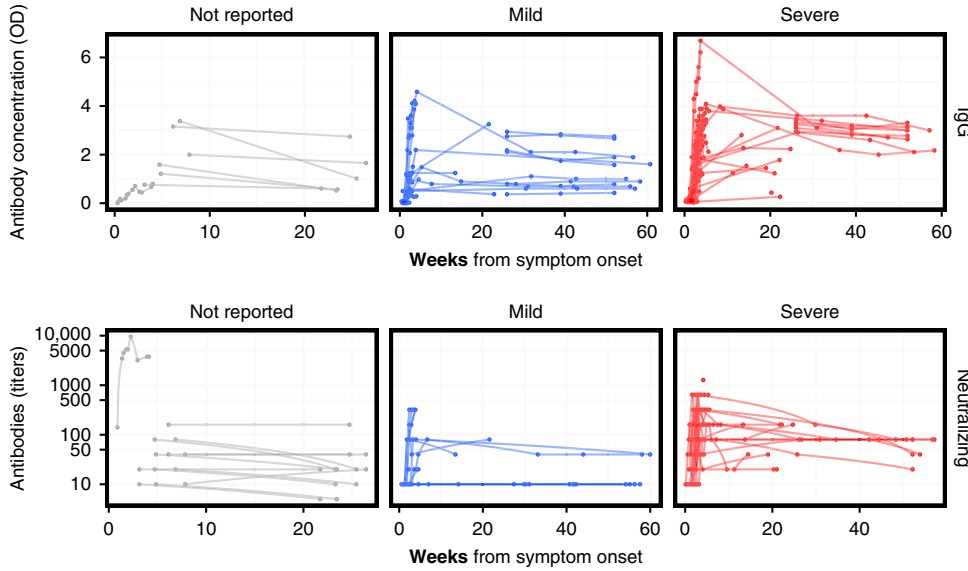

**Fig. 3 MERS-CoV antibody kinetics.** The top row shows the data for studies reporting IgG concentration in units of optical density, while the bottom row shows the data for studies reporting neutralizing antibodies in units of titers. The columns correspond to different severity categories. Each line corresponds to time series for an individual patient. Some studies reported titers that were lower than or greater than some threshold value; those are here plotted at those values (e.g., for ≥320, the value is assumed to be 320). Some studies may report the kinetics of different antibodies or using different assays (and different units) for the same patient. Note that while these are plotted on the same axis, values may not necessarily be comparable across studies within each panel, as each lab may have different assay conditions resulting in different scales. See Supplementary Figs. 3 and 4 for (more limited) the data on SARS-CoV and IgA. Colors reflect the severity categories: not reported (gray), mild (blue), and severe (red).

Interestingly, this study is one of the few to report illness as a function of dose of viral inoculum given in the challenge experiment, and suggests that those given higher doses (>$10^{1.2}$ TCD50) were more likely to experience cold (10/15) than those exposed to lower doses (<$10^{0.7}$ TCD50) (3/11).

Barrow et al.[56] found that lower proportions of individuals with high neutralizing titer experienced significant cold upon viral challenge than individuals with low titer.

Callow[12] characterized IgA, IgG, and neutralizing antibodies in sera and nasal washings from 33 volunteers before they were experimentally exposed to 229E HCoV. She found that multiple antibody responses were associated with reduced risk of infection, seroconversion, and symptomatic illness upon challenge. Individuals who seroconverted to the experimental viral exposure (defined as a rise in ELISA IgG serum antibodies) had significantly higher serum IgG, neutralizing antibodies, and nasal

**Table 2 Proportion of patients that had detectable antibodies, at different time points.**

| Strain | Weeks since symptom onset/infection | Asymptomatic | Mild symptoms | Severe symptoms |
|---|---|---|---|---|
| HCov-229E | 3 weeks | 20% (1 of 5) | 100% (10 of 10) | |
| MERS-CoV | 13 weeks | | 67% (10 of 15) | 95% (18 of 19) |
| | 26 weeks | | 67% (8 of 12) | 100% (10 of 10) |
| | 52 weeks | | 67% (8 of 12) | 100% (8 of 8) |

The time points were post symptom onset or infection, for HCoV-229E and MERS-CoV. The results for HCoV-229E are from Kraaijeveld et al.[51], and report the number of individuals experiencing symptoms of a given severity that had significant antibody rises post infection. The results for MERS-CoV are taken across studies in which a cutoff for the assay being used was provided, and gives the number of patients for which IgG levels were above the respective cutoffs, at different time points (also see Fig. 3). For example, at week 26, there were 12 patients with mild symptoms from studies that reported a cutoff, and 8 of those patients had IgG levels above that cutoff. MERS-CoV data were digitized from six studies[28,33,41,46,54,152].

IgA. Serum and mucosal IgA were associated with the duration of viral shedding post experimental infection, with those shedding for 5 days or more having statistically significantly less mucosal IgA than those shedding less than 5 days (0.6 ng/ml vs. 4.7 ng/ml, $P < 0.01$). Serum-neutralizing antibody was not found to be significantly associated with viral shedding duration. This study also showed protective associations of pre-infection serum-neutralizing antibody, serum IgG, and nasal IgA with clinical severity scores and nasal secretion weights (a measure of the severity of rhinorrhea symptoms).

Another prospective study[57] reported detection of pre-existing neutralizing antibodies among medical students who had virus isolation (67%, $n = 8/12$) or seroconversion to HCoV-229E (25%, $n = 3/12$). Pre-existing neutralizing antibodies were inversely associated with increases in neutralizing antibodies after reinfection, but were not associated with reinfection events that were determined by CF seroconversion.

Several studies exposed volunteers to two viral challenges, some months apart. Reed 9 rechallenged six volunteers who had been experimentally infected with HCoV-229E 8–12 months previously[9]. On the first challenge, all six developed symptoms and detectable viruses, and five of six experienced a significant rise in titer. In the second, zero of six experienced illness, detectable virus or significant rise in titer. Callow et al.[8] rechallenged volunteers with the same dose of HCoV-229E, 1 year apart. Of nine volunteers who were infected in the first exposure, 6 (67%) were infected in the second exposure. However, none of these individuals developed respiratory illness symptoms and they experienced a mean duration of detectable virus of 2 days compared to a mean of 5.6 in the initial challenge. Of note, these experimental doses may differ from the amounts of virus that people are exposed to in natural infections.

**Cross-reactivity and antigenic diversity**. We identified 82 papers as related to cross-reactivity and/or antigenic diversity (Supplementary Table 5). Of these studies, 59 were identified as highly important and were described in the text or tables, and data were digitized from 7 studies (Supplementary Data 2 and Supplementary Fig. 5). Figure 4 visually summarizes studies in this section.

Within the *Coronaviridae* family, the *Coronavirinae* subfamily includes four distinct genera. The Alphacoronaviruses include two major human coronaviruses, HCoV-229E and HCoV-NL63. Multiple HCoV-229E-like strains have also been characterized. The Betacoronaviruses are categorized into four lineages. Lineage A includes HCoV-OC43 and HCoV-HKU1, Lineage B includes SARS-CoV and SARS-CoV-2, Lineage C includes MERS-CoV and multiple bat coronaviruses, and Lineage D contains coronaviruses thus far only identified in bats. HCoV-OC43 and HCoV-229E are documented to cause common cold, while the more recent strains (HCoV-HKU1 and HCoV-NL63) infect both the upper and lower respiratory tract, resulting in more severe but rarely fatal disease[58]. Other CoVs have been associated with

human disease, including enteric disease in infants and zoonotic infections from livestock, but seem rare and are not described here[9,59–64].

Coronaviruses have four structural proteins: the spike protein (S), the nucleocapsid (N), the envelope protein (E), and the membrane protein (M)[65,66]. The S protein, which protrudes from the virus envelope, is immunodominant[50,67] and consists of two subunits: the S1 protein, which contains the receptor-binding domain (RBD) and the S2 protein, which mediates cell membrane fusion[68,69]. The nucleocapsid protein, which is also immunogenic, is smaller than S, lacks a glycosylation site, and induces antibodies sooner than to S during infection, making it an attractive protein for diagnostic assay design[70]. Sequence homology for the N and S of SARS-CoV to other Betacoronaviruses is 33–47 and 29%, respectively, while homology to Alphacoronaviruses is lower (25–29% homology to N and 23–25% for S)[70]. SARS-CoV-2 is most similar to SARS-CoV, harboring sequence homology of 90% in N and 76% in S followed by MERS-CoV (48% and 35%, respectively)[71]. Immunogenicity of other proteins were less studied. Studies in HCoV-229E[72] and SARS-CoV[73] suggest that M does have immunogenic epitopes, despite its relatively small size but with titers rising at times later than 21 days post infection, while responses to the E protein were rarely detected. In contrast, another study detected anti-M antibodies in over 80% of the individuals ($n = 58$) at 10 days post onset, while detection of anti-N and anti-S increased later[74]. The authors speculated these late detections as effects of their antigen-purification procedures where the N protein may not be able to exhibit its natural conformation as fragments of N were able to be detected earlier.

Natural and experimental infection studies in humans point to cross-reactivity within but minimal reactivity between endemic human Alpha- and Betacoronaviruses. Individuals experimentally inoculated with HCoV-229E and HCoV-229E-like strain LP experienced a >4-fold rise in neutralizing antibodies to both HCoV-229E and LP, while individuals inoculated with HCoV-OC43 did not[75]. While volunteers experimentally inoculated with HCoV-229E-like strains were protected against challenge with the homologous strain at 1 year ($n = 6/6$; none shed virus, showed symptoms, or had a rise in antibodies), volunteers experienced only partial protection against heterologous HCoV-229E-like strains ($n = 5/12$ protected). Increasing population immunity to HCoV-229E in the population was associated with less clinical disease upon challenge with HCoV-229E-like strains[9]. Studies of serological responses to HCoV N proteins point to cross-reactivity within Alpha-HCoVs (229E and NL63) and Beta-HCoVs (OC43 and HKU1), but not between Alpha- and Beta-HCoVs[76–79]. Consistent with observations in human challenge studies, children experiencing natural HCoV infections experience fourfold seroconversions to either HCoV-OC43 or HCoV-229E, but not both simultaneously[80]. However, a longitudinal study in newborns found that children seroconverted to either HCoV-NL63 or HCoV-229E but not both, although both viruses

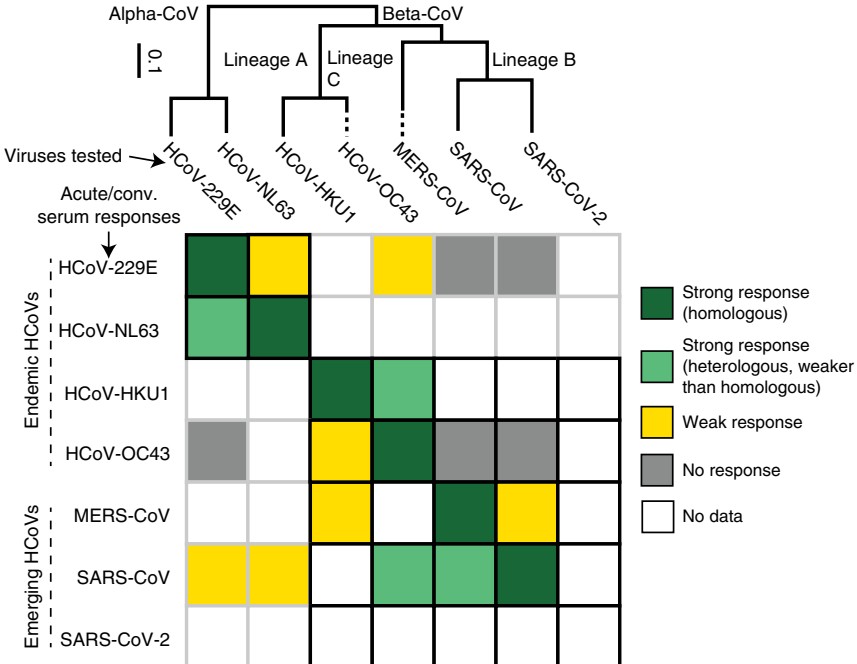

**Fig. 4 Antigenic and phylogenetic relationships among HCoVs.** Ordinal qualitative summary of the reactivity of antisera (rows) provided by individuals with confirmed infections with each human coronavirus against the panel of human coronaviruses (columns), shown in relation to their phylogeny[151]. Cell color indicates the magnitude of change in the antibody response (measured by neutralization assay, IFA, and/or enzyme-linked immunosorbent assays (ELISA) or western blot to N or S proteins) between acute and convalescent samples with dark green indicating strong, homologous response, light green indicating strong, heterologous response, yellow indicating weak response, and gray indicating no response. White boxes indicate that no data are available. The black grid indicates relationships between viruses of the same genera. Note that in some cases, cross-reactivity may be due to stimulation of immunity to previous infections (e.g., SARS-CoV serological responses toward HCoV-OC43), and in other cases, to relatedness of the viruses (SARS-CoV serological responses toward MERS-CoV).

are Alpha-HCoVs[81]. A later study in newborns followed from age 0 to 20 months[82] showed asymmetric interactions within Alpha-HCoVs and Beta-HCoVs: seroconversion to HCoV-NL63 was observed after HCoV-229E, but no recent HCoV-229E seroconversions had a recent infection by HCoV-NL63, suggesting that HCoV-NL63 provides at least short-term protective immunity against HCoV-229E. Similarly, HCoV-HKU1 seroconversion occurred prior to HCoV-OC43 but rarely after, suggesting that HCoV-OC43 protected against HCoV-HKU1.

Infection with endemic HCoVs produces little cross-reactivity to emerging CoVs SARS-CoV and MERS-CoV. Individuals experiencing natural infections with HCoV-OC43 or HCoV-229E did not have detectable antibodies in acute or convalescent samples against SARS-CoV (to N protein[83] by IFA or neutralization[45]). Healthy individuals with antibodies to HCoV-229E, HCoV-OC43, and other endemic HCoVs rarely had detectable antibodies that bound SARS-CoV-infected cells or SARS-CoV N protein[78,84,85]. HCoV-OC43 N and SARS-CoV N proteins have a subset of sites with shared homology, potentially explaining low-level false-positive results for N-based assays[86]. Blood donors in Southern China ($n =$ 152) and Saudi Arabia ($n =$ 130) did not have detectable binding (IFA) or neutralizing antibodies to either MERS-CoV or SARS-CoV[87,88]. Because children experience less severe disease during SARS-CoV infection than adults, it was hypothesized that childhood vaccination with non-CoVs provided cross-protection against SARS-CoV. However, binding and neutralizing antibodies and T-cell responses induced by routine childhood vaccinations (AMPV, BCG, DPT, HBV, HIB, JEV, MMRV [MV and RV], OPV, PI, SV, and VV (varicella vaccine)) did not cross-react with SARS-CoV in experimentally inoculated mice[89].

Emerging HCoVs can induce cross-reactive binding antibodies toward endemic and other emerging HCoVs. SARS patients often experience a > 4-fold rise in antibodies to HCoV-229E, HCoV-NL63, and/or HCoV-OC43 in paired acute/convalescent samples ($n =$ 12/20)[45]. In one study, the number of SARS patients who experienced a greater than fourfold rise in binding antibodies was greater to HCoV-OC43 ($n =$ 10/11) than HCoV-229E ($n =$ 5/11)[84]. Some SARS patients also showed a rise in antibodies to HCoV-229E and HCoV-OC43 N protein[84] and HCoV-NL63[76]. Among SARS-CoV patients ($n =$ 28), 60% had detectable IFA titers to MERS-CoV and 25% had anti-MERS-CoV- neutralizing antibodies. A subset with available paired samples experienced seroconversion to HCoV-OC43, but limited seroconversion to Alpha-CoVs. In the same study, animal handlers at a wildlife market in Guangzhou ($n =$ 94) with low-level prevalence of antibodies to SARS-CoV (13.8% by IFA measuring an antibody bound to infected cells, 4.3% by NT) had detectable antibodies toward MERS-CoV (2.2% by IFA)[87]. A follow-up study found that the cross-reactivity between SARS-CoV and MERS-CoV was unlikely to be due to similarity in the RBD, as monoclonal antibodies (mAbs) raised to SARS-CoV RBD did not bind MERS-CoV RBD or neutralize MERS-CoV even at high concentrations[90]. MERS-CoV may produce less cross-reactivity against SARS-CoV. A subset of slaughterhouse workers in Saudi Arabia (a setting seen as potentially high risk for MERS-CoV exposure) had antibodies to MERS-CoV by IFA as well as to endemic HCoVs, but none had reactivity to SARS-CoV spike protein. MERS-CoV patients were observed to have low-level cross-reactivity to SARS-CoV[88] and in one study HCoV-HKU1 N protein[76]. Because of this cross-reactivity, researchers have developed diagnostic assays with truncated SARS-CoV S, N, and M proteins[91–94]. A study with longitudinal follow-up suggests that the titers may indeed reflect stimulation of pre-existing antibodies from past infections as avidity of IgG

antibodies to HCoV-OC43 and/or HCoV-229E in two of their SARS patients was high early on and remained high, while the avidity against SARS-CoV was initially low[24]. See Fig. 4 for associations between homologous and heterologous titers.

There is some evidence for antigenic evolution in the receptor-binding domain of emerging CoVs. A study of SARS-CoV strains from the zoonotic phase (palm civets and bats) to early and late in the SARS epidemic revealed that some escaped neutralization by mAbs targeting the spike RBD of SARS-CoV[95–97]. Sera from BALB/c mice immunized with full-length S protein from civet strains were ineffective against human SARS-CoV and vice versa[98]. Despite significant cross-reactions between mAbs against conformational epitopes of RBD with multiple mutational differences, single mutations were shown to disrupt neutraliz-ability[99]. In contrast, mAbs targeting regions critical for fusion and entry on the S2 protein are immunogenic and can broadly neutralize SARS-CoV strains[65,100]. A recent study showed reduced binding of mAbs from SARS patients to the RBD of SARS-CoV-2, especially those that blocked binding to the ACE2 receptor. The only mAb potently bound to SARS-CoV-2 RBD protein did not compete with the RBD for binding to the ACE2 receptor, suggesting that it bound a different, conserved site on the protein[101]. Similar studies were conducted to study changes in MERS-CoV. Five recombinant RBD proteins were constructed with mutations detected from MERS-CoV strains isolated in humans (2012–2015) and camels[102]. These RBDs maintained functionality and induced potent neutralizing antibodies. When residues in their receptor-binding motifs were mutated to evade neutralization, cross-reactivity persisted, but binding affinity to DPP-4 (the main receptor for MERS-CoV) was lost, suggesting limited antigenic escape for MERS-CoV. A study of MERS-CoV isolates with distinct amino acid differences in the S and other replication proteins found differences in replication kinetics, but it is not clear that these were attributable to differences in S[77].

**Immunopathogenesis**. In the initial review, 44 papers were identified as related to immunopathogenesis. Of these, we found 26 sufficiently relevant to review in Supplementary Table 6, and 16 of those are detailed below and/or summarized in Fig. 5.

Antibody-dependent enhancement or other antibody-mediated immunopathogenesis may be possible. Antibody-dependent enhancement (ADE), where pre-existing antibodies increase pathogenicity by facilitating viral entry into cells, a long hypothesized explanation for severe dengue infections, has been hypothesized to play a role in coronavirus pathogenesis specifically in patients who seroconverted early in SARS-CoV infection[17,29,103]. In support of a priming role from pre-existing antibodies against endemic strains is the observation that older patients infected with SARS-CoV appeared to mount an earlier immune response with higher titers than younger patients[103]. Authors from another study pointed toward enhancement from antibodies within a single-infection episode[17]. Nasopharyngeal viral load increased in the first week and declined thereafter, but clinical worsening was seen in many of the patients at week 2, with virus shedding in stool and urine observed toward the end[17,29]. Many manifested with additional new lesions as original lesions improved. The timing of the appearance of new lesions correlated with IgG seroconversion and declined viral load, suggesting that pathology post week 1 was driven by the immune response rather than by uncontrolled viral replication. Others have argued that ADE due to neutralizing antibodies is unlikely as treatment of SARS-CoV with convalescent serum did not result in adverse effects[103].

Immunopathogenesis has been observed in a coronavirus of cats, feline infectious peritonitis virus (FIPV). Kittens passively immunized with serum-containing antibodies to FIPV developed more rapid disease than controls not passively immunized[104]. The authors noted similarity to dengue in humans where infants with mid-levels of maternal antibody to dengue (compared to high or low levels of maternal antibody) experience increased risk of disease[105]. The one study that we found that addressed this characterized endemic HCoV in infants[106]. They observed the highest lower respiratory tract infection burden (7.8%) at 6–23 months of age (1.5% at <6 months, close to none at 2–5 years), while upper respiratory tract infection burden was close to uniform[106]. The heightened disease burden after maternal immunity waning to medium levels in the lower tract, and the absence of such observation in the upper tract where antibodies do not circulate, subtly supports the possibility of ADE in HCoVs.

Controlled in vitro experiments have explored the possible enhancing action of antibodies for HCoV infections. A series of studies by Yip et al. demonstrated that opsonization of anti-spike antibodies allowed SARS-CoV to enter non-ACE2-expressing immune cells that bear Fc-γ-RII (CD32)[107,108]. Though replication is observed after entry, the virus does neither exit nor alter the expression of proinflammatory immune mediators (CCL2/MCP-1, CCL3/MIP-1α, CXCL10/IP-10, and TNF-α) and apoptosis-inducing ligands (FasL)[109]. This contrasts with the effects in nonhuman primate studies where endocytosis into macrophages stimulated inflammation, which in turn drives severe lung injuries[110]. A study by another group in the human promonocyte cell line (HL-CZ), which expresses both ACE2 and Fc-γ-RII, demonstrated increased infectivity and virus-induced apoptosis when anti-SARS-CoV sera from patients were added at 100- to 2000-fold dilutions, while at higher concentrations neutralization occurred[111]. Upon infection, TNF-α, IL-4, and IL-6 expressions heightened, while IL-3 and IL-1β only appeared in trace amounts. The difference could have resulted from the cell line differences or the set of mediators assessed[109]. Regarding regions of the spike protein that may induce antibodies with enhancement effects, both anti-S1a and anti-S1b mAbs showed mild-to-moderate effects[111]. Only one particular anti-S1b clone showed neutralization. No effect was seen for anti-N mAbs. There is limited evidence that these mechanisms are causal to inflammatory gene-expression differences among patients with differing severities[112].

While the link between antibody presence and enhanced severity via infection of immune cells remains unclear, some have suspected a role for autoreactive responses in increased severity of illness. Prolonged tissue destruction can increase presentation of host proteins to T- or B cells and result in an adaptive response against self-, i.e., epitope spreading[58]. Increases in anticardiolipin antibodies were reported in 33.9% of 62 post-SARS osteonecrosis patients, but the lack of a comparison group of post-SARS patients without osteonecrosis meant that the link was incon-clusive[113]. Anti-S2 IgG antibodies targeting uninfected lung epithelial cells (A549) were detected in SARS-CoV patients 20 days post symptom onset[114], a reactivity not seen in serum from healthy individuals and non-SARS-CoV pneumonia patients. Complement inactivation only showed a cytotoxic effect when IgG was present/unbounded. The presence of anti-S2 antibodies also increased the binding of immune cells (PBMCs) to A549 cells treated with IFN-γ, replicating the conditions under which a cytokine storm would be observed. A separate study demonstrated colocalization of anti-S2 antibodies collected from serum of SARS patients ≥50 days post fever onset with annexin A2 and immunoprecipitated annexin A2 on A549 cell surfaces. Elevated expression of annexin A2 on the surface can be stimulated by IL-6 and IFN-γ, both cytokines induced by SARS-CoV, which in turn increases binding of anti-S2 antibodies to the cell. However, its pathogenic role was not explored[115].

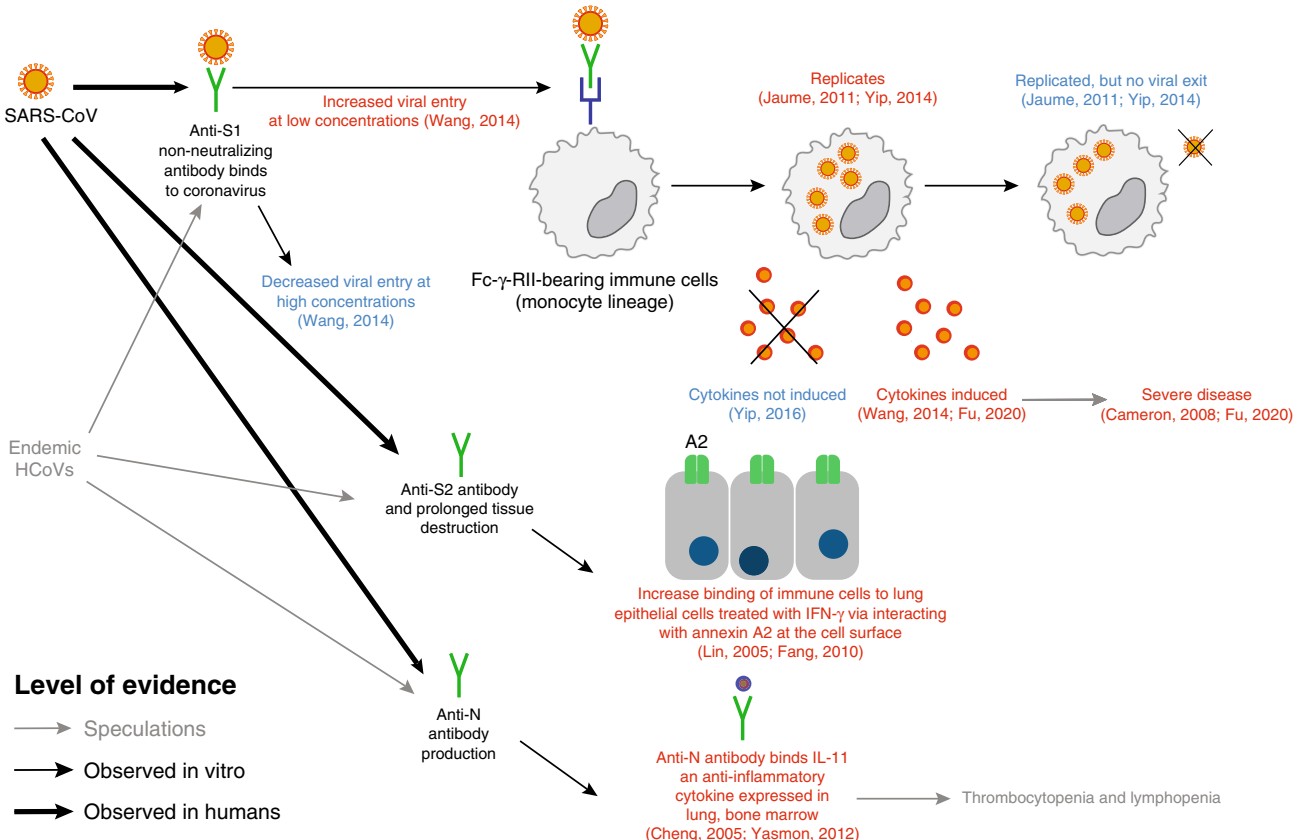

**Fig. 5 Evidence supporting/contradicting SARS-CoV antibody-related pathogenesis. Supporting evidence given in red and contradicting evidence in blue.** Weaker evidence (speculation in discussions) shown in gray, evidence from in vitro studies in thin lines, and evidence observed in humans in thick lines. Anti-S1 antibodies triggered upon infection may facilitate entry into immune cells at later stages of the infection if concentration is low. Replication happens but no virus is released. Consequential induction of cytokines is inconclusive, but if they occur, they are associated with severe disease. Roles of anti-S2 and anti-N antibodies are supported by binding observations.

Alternatively, similarity between viral and host epitopes (molecular mimicry) can generate cross-reactive antibodies. In mice[116] and humans[117], there is evidence of anti-N antibodies that cross-react with IL-11, an anti-inflammatory cytokine expressed in many tissues, including the lung and bone marrow. The authors suggest that high anti-N antibodies induced relatively early during infection may be involved in the thrombocytopenia and lymphopenia observed early in SARS-CoV infection[116].

**Population seroprevalence**. From the paper review, 68 papers were classified as having data on age and seroprevalence or seroincidence, of which 20 studies were confirmed on further review, and 14 had digitizable data. The age range across studies was 0 to ≥65, while the sample size of studies ranged from 69 to 19,974 (82–6400 for studies of endemic HCoVs). Supplementary Table 7 and Supplementary Data 3 contain details of the studies.

For endemic coronaviruses, seroprevalence rises sharply in childhood, with little to no change in seroprevalence by age among adults. While the exact dynamics from 6 months to 20 years vary between studies, the general trend remains consistent. Figure 6 displays HCoV age-seroprevalence curves for six papers with digitizable data on seroprevalence by age, with the panels representing the four major endemic strains[118–123]. Trends from papers not displayed in the figure are largely the same[78,81,124,125]. Two studies show a marked decline in seroprevalence with age above 40[78,119].

The force of infection of endemic coronavirus strains is high, and the age at first infection low, but variable across studies. Simple catalytic models fit to the digitized data predict median total force of infection across studies of 0.21 (95% CI 0.09, 0.40) among immunologically naive individuals (see Fig. 6 for fitted curves for each study and strain, and Supplementary Table 8 for estimates by study and strain), corresponding to an average age at first infection with any strain of 4.8 years (95% CI 2.5, 11.2). A cohort of 25 infants followed from birth for an average of 2 years experienced annual strain-specific incidence rates from 0.12 to 0.70[81,82]. Zhou et al.'s serosurvey measured IgG and IgM separately, and showed very different patterns with age[123]. While IgG seroprevalence rose to high levels by age 10 and remained high into the adult population, IgM seroprevalence declined to zero for all individuals aged 14 and older. The authors interpreted this as evidence that first infections occurred before the age of 14 for all strains.

Available datasets are more sparse for incidence, and the patterns are inconsistent between studies[118,126–128]. There was an increasing trend in seroincidence with age during an outbreak for HCoV-229E in Tecumseh, Michigan[118], but a similar analysis of HCoV-OC43 seroincidence in the same outbreak showed no such increase[126]. Longitudinal follow-up of ten families in Seattle found a lower rate of seroconversions among adults compared to children[127]. A comparison of a cohort of 21–40-year olds and a cohort of ≥65 years old found no clear difference in incidence between the two age groups, as measured by seroconversion or PCR-confirmed, HCoV-associated respiratory illness[128]. Finally, observational studies have shown evidence of coronavirus-associated respiratory illnesses in elderly populations[129–131] and across all ages[132]. Supplementary Fig. 6 displays age-incidence curves for the four papers with digitizable data on incidence by age.

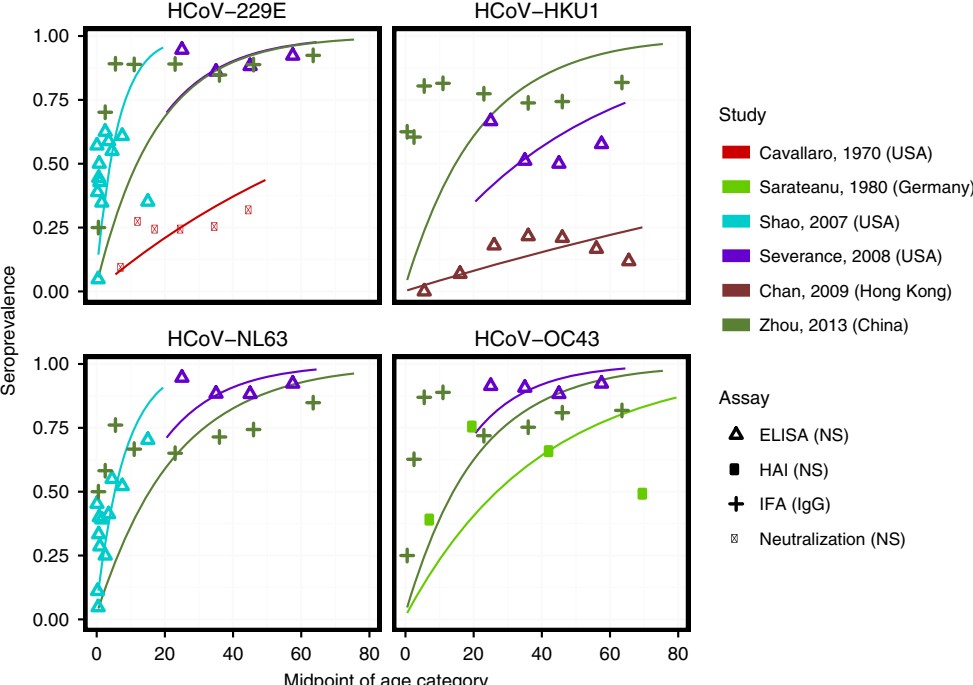

**Fig. 6 Age-seroprevalence curves on endemic HCoV.** The color denotes the study, and the point type denotes the assay and antibody measured. The data from 134 are averaged over two serosurveys conducted in 1975 and 1976. Points are the observed proportion seropositive in each age group, while lines are predicted age-seroprevalence curves from catalytic models fit to each study and strain separately.

The rapid rise in seroprevalence with age observed in the literature suggests a force of infection (and thus proportion seropositive to all serotypes at older ages) high enough to preclude significant rates of coronavirus infection among older adults if each serotype confers lifelong, complete homologous immunity (Fig. 7, red line). On the other hand, short-term, complete immunity, lifelong partial immunity, and the existence of multiple genotypes within a single serotype with limited cross-immunity could explain the age-seroprevalence and age-incidence curves seen in the literature (Fig. 7).

The review gave rise to several observations that were either consistent or conflicting across papers. Studies that enrolled children less than 6 months detected loss of maternal antibodies, representing a preliminary line of defense for newborn children[81,82,85,125]. Most studies showed no discernible difference in age-seroprevalence trends or in the overall seroprevalence by strain. Gao et al.[78] found that seroprevalence of HCoV-229E and HCoV-HKU1 was significantly lower than seroprevalence of HCoV-OC43 and HCoV-NL63. In addition, Chan et al.[122] found seroprevalence to HCoV-HKU1 to be low (21.6% in 31–40-year olds) in Hong Kong, and stated that this was expected from the low rates of HCoV-HKU1 among respiratory illnesses. Finally, most studies measured the presence of binding antibodies in the blood. Of the studies in Fig. 6, only Cavallaro and Monto[118] (red) measured seropositivity using a neutralization assay. That the seroprevalence is markedly lower in that study could indicate lower prevalences of neutralizing antibodies, lower sensitivity of neutralizing versus binding assays, or lack of correlation between neutralizing and binding antibodies against endemic coronaviruses.

Regarding the novel coronaviruses, serosurveys of SARS-CoV confirmed that the rate of asymptomatic or subclinical infection was very low[133]. Asymptomatic and subclinical rates of MERS-CoV are generally higher, but available serosurveys lack the power to draw strong inference about age trends. A large serosurvey conducted across Saudi Arabia found that the age of seropositive individuals was significantly lower than the age of

clinical cases[134], while another conducted across multiple countries in Africa and Asia found no trend in seroprevalence with age[135]. Studies of risk factors within camel workers have either not addressed age as a risk factor[136,137] or not found an association[138].

## Discussion

We have presented a broad, comprehensive review of multiple aspects of the literature on antibody immunity to coronaviruses. We identified a number of key findings. The median time to detection was the shortest for SARS-CoV-2 (11.0 days; IQR 7.0–14.0 days), followed by SARS-CoV (13.5 days; IQR 10.0–18.0 days) and MERS-CoV (15.0 days; IQR 12.0–18.0 days). Most long-term studies found that SARS-CoV and MERS-CoV IgG waned over time (typically detectable up to at least a year), while others found detectable levels of IgG 3 years post symptom onset. Antibody kinetics varied across the severity gradient, with antibodies remaining detectable longer after illness with more severe symptoms. Human challenge studies with HCoV indicate that serum and mucosal immune responses (serum IgG, IgA, neutralizing titer, and mucosal IgA) provide possible correlates of protection from infection and disease. However, repeat human challenge experiments with single HCoV suggest that individuals can be infected with the same HCoV 1 year after the first challenge, though perhaps experiencing lower severity. There is cross-reactivity within but minimal reactivity between Alpha- and Beta-CoVs. While endemic HCoVs rarely induce cross-reactive antibodies against emerging HCoVs, SARS-CoV and MERS-CoV stimulate antibodies induced by prior HCoV infections. Multiple mechanisms for immunopathology have been suggested, but no strong causal evidence exists. The extent to which antibody responses can in some circumstances contribute to disease severity is not known. Seroprevalence with the four major endemic HCoV strains rose rapidly during childhood and remained high in adults. Median age at first infection with any

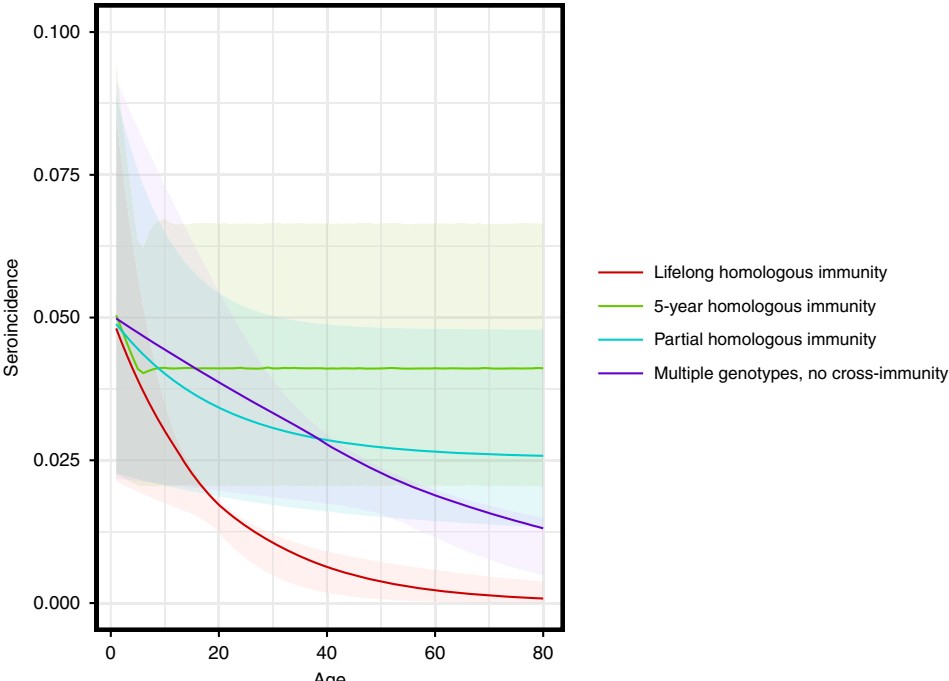

**Fig. 7 Age-seroincidence curves under four models of coronavirus immunity.** Each curve displays the age seroincidence for a single endemic strain. The four hypothetical models of coronavirus immunity are (1) infection grants lifelong, complete homologous immunity (red), (2) infection grants complete homologous immunity for 5 years, then reversion to complete susceptibility (green), (3) infection grants lifelong homologous immunity at 50% efficacy (cyan), and (4) the strain consists of four antigenically diverse genotypes, each granting complete homologous immunity within genotype, but no cross-protection between genotypes (blue). Confidence bands represent uncertainty in the force of infection as estimated from the seroprevalence data.

strain was 4.8 years (95% CI 2.5–11.2 years). There was no clear trend in seroincidence with age, and many studies have demonstrated incidence of coronavirus infections in elderly populations. These results suggest a measurable impact of immunity to coronaviruses on future risk, but this protection may be transient.

We have suggested a set of search terms to identify relevant work, and others may expand on this search. Due to the speed of new research being produced, we limited our systematic review of SARS-CoV-2 papers to before March 20, 2020. Further evidence on immune responses to SARS-CoV-2 is likely contained in work that has since been published. To remain human-focused, we excluded from the review animal studies and studies of animal CoVs, but this literature is likely relevant to some areas of the review. We limited the scope of the review to antibody-mediated immunity, meaning that understanding of some areas of the review may be incomplete. Other aspects of immunity, including the innate and cellular responses, have been shown to be important in conferring protection at reexposure. Finally, digitized data could be affected by publication bias or other selection bias, meaning that it might not be representative of data from all studies. The aim of the pooled analyses was to summarize the array of studies rather than formally explore hypotheses, and as a result, the models used are simple. For example, in estimating force of infection from age-stratified seroprevalence data, we did not consider other features of a model, such as assay sensitivity, time- and age-varying force of infections, seroreversion, and cross-reactivity between strains, that might better explain patterns seen in the data. Albeit the limitations, there exist multiple implications for the ongoing SARS-CoV-2 pandemic responses and future outlooks.

There is a need for development of serological assays with high sensitivity and specificity to understand the extent of infection in populations. Evidence from previous emerging CoVs suggests

that false negativity is likely to result from waning of antibody levels. As most studies on antibody kinetics are on symptomatic patients, the kinetics in subclinical infections, a significant proportion of SARS-CoV-2 infections[139], remains a key gap in the literature. Serological surveys to describe the extent of infection in particular populations should account for the dynamics of antibody and the potential for infections associated with different severities of illness to have different antibody responses in their analysis. Indications that less severe illnesses are associated with reduced antibody responses suggest that mild cases may pose challenges to serological assays.

On the other end, the proposed policy of immunity passports to allow individuals who are presumed to be immune to return to work once social distancing measures are relaxed requires highly specific serological tests to mitigate adverse outcomes. Evidence from past emerging HCoVs suggests low false positivity from cross-reaction with endemic HCoVs. However, antibody titers do not necessarily translate to immunity. Challenge studies indicate multiple candidates that may serve as correlates of protection, including serum and mucosal measures; however, these will need specific evaluation for SARS-CoV-2. The knowledge gap in correlates of protection and their durability must be filled before immunity passports are safe for general use.

Numerous vaccines of varying antigenic compositions are currently under development[140], and polyclonal antibodies from SARS-CoV-2-recovered patients are being evaluated for treatment[141]. In this review, we have summarized concerns of theoretical dangers from antibodies in the immunopathogenesis of COVID-19 diseases. These concerns should not hinder the urgent development of vaccines and therapeutics; however, these hypothetical risks must be squarely addressed. Research in animals, in vitro, and in humans should be done on immunopathogenesis of SARS-CoV-2 as an integral component of the development of antibody therapeutics and vaccines. Compared to other disease

systems (e.g., dengue), where risk from antibody-mediated immunopathogenesis is well supported with laboratory, clinical, and epidemiological data, this potential for immunopathogenesis in SARS-CoV-2 infections remains speculative[142–145]. Though evidence for immunopathogenesis remains limited[146], it will be important to design human trials to provide valid evidence of a positive effect and detect negative effects should they occur. There are concerns with antibody treatment, where patients are already infected and ill-pose lesser concerns. Preventive measures, including vaccines or passively administered antibodies meant to lower risk, will require closer attention. Serological responses in vaccinees should be evaluated to detect potential immuno-pathogenic effects. Human vaccine trials might start with older individuals, who per infection are at higher risk of experiencing severe illness, and as such have a potential benefit-to-risk ratio from vaccines that is much higher than younger individuals. Consideration could also be given to post-licensing surveillance systems designed to detect the remote risks of waning immunity and subsequent adverse events due to vaccine-mediated immunopathogenesis.

Key questions for the long-term dynamics of SARS-CoV-2 include whether it is likely to be eradicated, whether widespread immunity will lead to transient decreases in incidence, and how long it will take for another large outbreak to occur (Table 1). Modeling studies will be crucial to answer these questions, but they will rely on an understanding of immune responses to SARS-CoV-2 and interactions with other coronaviruses to make relevant predictions. Analysis of datasets, specifically of serosurveys, needs to account for the kinetics of waning immunity and effects of imperfect assays to draw meaningful inferences.

Finally, we note that the best evidence for immune responses to SARS-CoV-2 will come from studies of the virus itself. Such studies are being performed and reported with unprecedented speed, but in the absence of firm evidence, many are turning to the existing coronaviruses as a model. To that end, we have produced this review to aid researchers in understanding the scope of the evidence for immune responses to these coronaviruses.

## Methods

**Search strategy and selection criteria**. We conducted searches of the PubMed database on March 20, 2020 using the search term "coronavirus" and each of the following terms or phrases: "serolog*", "serop*", "cross reactivity", "antibod* human coronavirus", and "complement fixation". We also searched for each of these terms with the search terms "SARS" and "MERS". Articles were included in the search, regardless of publication date. Articles included electronic, ahead-of-print publications available in the PubMed database. We did not attempt to contact authors to obtain unpublished data. Each abstract was reviewed by two reviewers. Articles that were not considered applicable were excluded (Supplementary Fig. 1). We additionally identified relevant articles from the references of other articles identified by our searches.

**Assessment**. Abstracts were classified as (1) study of antibody immune responses to human coronavirus, (2) study of human coronavirus, but not about antibody immunity, (3) study in nonhumans or basic biological study of virus–antibody interactions not characterizing actual humans, (4) study in humans not about coronavirus, (5) in another language, (6) duplicate of another record, or (7) dead link/bad return/noise. Disagreement between two reviewers on classification was resolved by a third reviewer. Papers classified in the first category (study of antibody immune responses to human coronavirus) were reviewed in their entirety and classified by one reviewer as relevant to one or more of the following: (1) antibody kinetics, (2) correlates of protection, (3) antigenic diversity and cross-reactivity, (4) immunopathogenesis, and/or (5) population seroprevalence or incidence. Papers that were not considered relevant to any of the preceding categories were excluded.

**Data abstraction**. Data were digitized from papers according to predefined criteria for each area of focus, related to accessibility, utility in answering key questions, and suitability for pooled analyses.

For antibody kinetics and the association of antibody responses with clinical severity, we shortlisted papers that met the following criteria for data digitization:

(i) antibody responses were provided for at least two distinct points in time and were relative to a point of infection or symptom onset, or (ii) explicitly discussed severity of symptoms in relation to antibody response. Data could consist of multiple measures of antibody kinetics: serological responses for individual patients (either quantified antibody levels or binary metric of seropositivity), and (cumulative) proportion of a group of patients that was seropositive or had seroconverted at different time points. Most studies used more than one assay and targeted more than one antibody; in these cases, we digitized all the data provided across assays and antibody types. For the pooled analysis, we excluded studies that summarized antibody responses across patients, but we nonetheless discussed their main findings. Where possible, severity associated with different (groups of) patients was extracted and standardized (Supplementary Table 3). Finally, we also digitized cut-off points for the assays when given to define limits of detection or thresholds for positivity, and refer to these as limits of detection. When not explicitly stated, but a category defined as less than some value exists, we assumed the value to be the limit of detection (e.g., cutoff of 1:10 is assumed when "<1:10" was present).

Studies on correlates of protection targeted for inclusion required a defined exposure, pre-existing antibody level (either antibody concentrations and/or serostatus) and outcomes as either virologically confirmed or serologically confirmed infection.

For cross-protection and antigenic diversity, instances of human infections with a particular HCoV and titers against itself and the others were summarized. Acute, convalescent, and fold rise in titers were digitized as brackets of possible values along with the number of individuals associated with those data points (e.g., a titer reported as 1:160 with the next serial dilution tested at 1:320 could take a value from 1:160 to 1:320). For MERS-CoV and SARS-CoV, acute titers were assumed to be the lowest reported in that study (either <1:10 or <1:20) as prior exposure was unlikely (Leung et al.[133]; Degnah et al.[135]). If not reported, fold rises were calculated using lower ends of both time points. If measurements included the lowest reported, we assumed a titer of five to avoid having a zero as the denominator. All data points are accompanied by the type of test/assay performed.

For population seroprevalence, we determined papers that reported the number of positive tests out of the number sampled in at least two age groups in a population (by seroconversion with/without symptoms or PCR-confirmed symptomatic infections) if sampling was performed independently of symptoms. Data extracted from the text, tables, or figures include strain/virus tested for; whether seropositivity, seroconversion, or incidence was measured; time period of the study for seroincidence studies; assay type; target antigen; cut point for defining seropositivity/seroconversion; bounds of age category; number of samples; number of positive samples. In plotting the data, if the upper bound of the highest age category could not be identified, we assumed that it was 20 years above the lower bound based on inspection of the highest age categories in other studies.

For diagnostic serological assays and immunopathogenesis, we compiled and summarized findings from the categories without data extraction. Data extracted for correlates of protection and antigenic diversity were summarized and visualized without the attempt to draw a pooled conclusion.

**Pooled analyses**. We conducted pooled analyses in two areas.

To analyze the antibody kinetics and association of antibody responses with clinical severity, we summarized the approximate time to detection for three sets of studies. The first set includes studies with a cumulative number of positive cases over time. Each additional positive case added to the cumulative curve was assumed to be a new detected case at that time point. This is approximate only because cumulative curves sometimes decline (Supplementary Fig. 2), indicating that some participants were lost to follow-up. Second, we identified the data that described either seropositivity as a function of time, or measures of antibodies with a reported cutoff. Data were included if (i) the first seropositive time point was within 3 weeks of illness onset, or (ii) the first estimate(s) were negative, but a subsequent measurement within 3 weeks of the first was positive. Third, if a study only reported times to seroconversion, those values were used.

We estimated the annual force of infection from age-stratified seroprevalence data, assuming that individuals acquire infection with a single, strain-specific force of infection that is constant in time using standard method 18, and that once an individual becomes seropositive, they remain seropositive for the remainder of their lifetime. We estimated the total force of infection (FOI) from the four major endemic strains by drawing 100,000 bootstrap samples of four strain-specific FOIs and calculating the sum across strains. Age at first infection is given by the inverse of the total FOI. Finally, we visually compared age-incidence trends to curves generated by catalytic models with four proposed mechanisms for CoV immunity (see "Supplementary Appendix" for details).

Hypothetical age-seroincidence curves using different models for endemic HCoV immunity were built using extensions of the simple catalytic model in which each individual is exposed to a constant force of infection $\lambda$, and the probability of being seropositive at age a is given by $S(a) = 1 - e^{-\lambda a}$, and the incidence rate from age a to a + 1 is given by $I(a) = e^{-\lambda a} - e^{-\lambda(a+1)}$. For example, if the first infection grants partial homologous immunity, reducing the hazard of infection by a factor $\rho < 1$, the incidence rate from age a to a + 1 is given by $I(a) = e^{-\lambda a}(1 - e^{-\lambda}) + (1 - e^{-\lambda a})(1 - e^{-\rho \lambda a})$.

## Disclaimer

Material has been reviewed by the Walter Reed Army Institute of Research. There is no objection to its presentation and/or publication. The opinions or assertions contained herein are the private views of the author, and are not to be construed as official, or as reflecting the true views of the Department of the Army or the Department of Defense.

**Reporting summary**. Further information on research design is available in the Nature Research Reporting Summary linked to this article.

## Data availability

The authors declare that all data generated or analyzed during this study are included in this published article and its Supplementary Information files.

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

## Acknowledgements

This research was supported [in part] by the Intramural Research Program of the National Human Genome Research Institute, National Institutes of Health. D.A.T.C. was supported by the US National Institutes of Health award number R01-AI114703-01.

A.P.W. is funded by a Career Award at the Scientific Interface from the Burroughs Wellcome Fund and by the National Library of Medicine of the National Institutes of Health under award number DP2LM013102. D.A.T.C. and A.P.W. were also supported by a grant from the Burroughs Wellcome Fund (1021212). I.R.B. was supported by the John A Watson Faculty Scholar fellowship. We thank Celeste Dale and Maxwell Hogshead for assistance in data digitization.

## Author contributions

A.T.H., B.G.C., M.D.T.H., B.Y., L.C.K., S.M.R., B.A.B., C.A.M., B.D.S., and D.A.T.C. conceived and designed the study. A.T.H., B.G.C., M.D.T.H., B.Y., L.C.K., S.M.R., B.A.B., C.A.M., B.D.S., L.T.S., V.E., I.R.B., J.L., H.S., D.S.B., A.W., and D.A.T.C. contributed to the data acquisition and collation. A.T.H., B.G.C., M.D.T.H., B.Y., and L.C.K. contributed to analysis of pooled data. All authors had substantial contributions to the interpretation of the data, writing, and review of the final paper.

## Competing interests

The authors declare no competing interests.
