## [Peer Review File · Nature Communications]

Reviewers' Comments:

Reviewer #1:

Remarks to the Author:

The review discusses our current knowledge on Coronavirus antibody responses in the humans in relation to kinetics, cross reactivity and potential immunopathogenesis. The review is timely and important to add to our current understanding of immune responses to the current coronavirus SARS-CoV-2 which is needed for the pandemic management and release of lockdown measures, as well as counter measures and vaccine development; and thus, it is of interest to a wide community.

However, a main concern is: since the review focuses on the antibody responses to coronavirus, the term antibody has not been included in the search terms, which could have lead to missing some relevant research articles.

* while the term antibody was missing from the search terms, the authors looked for "complement fixation", why was that specifically chosen? specially it is not the only antibody effector function.

*Some Studies on antibody responses to SARS-CoV-2, kinetics of antibody isotypes, and cross reactivity were available online within the selected timeframe were not included in the review and could add up to the limited information available on SARS-CoV-2 immune responses at the time when the review was conducted, for example: <https://doi.org/10.1101/2020.03.18.20038059> (PMID: 32267220) and <https://doi.org/10.1101/2020.03.18.20038018> (PMID: 32350462) specially that the authors have referenced preprints (ref 35).

* Where there any studies describing antibody responses against other structural proteins apart from the S and N proteins? This needs to be discussed in the manuscript and their role in cross-reactivity (specially that the authors mentioned the use of a truncated M protein in page 13 of the manuscript, with no previous mention to any responses against this protein)

* Would be useful to show the degree of homology of the four structural proteins among the 7 known coronaviruses in a table format. relevant for figure 4.

Minor comments:

*Table 1 legend: reference to Figure 4 should be Figure 3

*Figure 2 legend: MERS-CoV (left row) correct to (upper row)

*Figure S5: upper panel is missing the x-axis numbering

*page 13, first paragraph, line5: IFA is used once and IF is used in the second time, please use one form for consistency and check this and other abbreviations throughout the manuscript.

*page 13, first paragraph, line5: which type of IFA showed the cross reactivity? is it a whole virus or a recombinant protein and if so which protein? please mention for clarification. because in the following sentence the authors jump to RBD similarity which is unlikely to be a reason for the observed cross reactivity.

*page 13, first paragraph, line11: correct pandemic HCoV to endemic HCoV

Reviewer #2:

Remarks to the Author:

This is a fantastic review by Huang, Garcia-Carreras, Hitchings, Yang, Katzelnick and colleagues. I found the review to be very informative and well-written. The authors identified a number of critical papers that I was not aware of and I think that many people in the field will find this review useful.

Some points to consider:

1. The review is focused on antibodies, but the authors should at least mention the involvement of cellular responses. For example, it is quite possible that T cell immunity is involved with cross-coronavirus protection. The authors should consider adding a small section that discusses immune responses other than antibodies. Consider editing page 5 sentence 'an exposure to a pathogens generates an immune response that changes over time and between individuals (antibody kinetics)'. Change 'immune response' to 'antibody response' since that what is examined in this review.
2. In the review, the authors correctly predicted that a lot of new SARS-CoV-2 data would become available before this is published. The authors might want to add a final paragraph that summarizes key finding published since their March literature searches, including papers describing the kinetics of IgG/IgM SARS-CoV-2 responses and development of diagnostics and therapeutic antibodies.
3. Edit Figure 2 X axis to be more precise 'Time between symptom onset and detection'
4. Figure 3: what is each line and data point. Describe in text and legend.
5. Page 7: HAI is not an antibody binding assay.
6. The authors should consider better describing the limitations of challenge studies. Very high doses of virus are used for these studies and could explain why there seems to be incomplete immunity in rechallenger studies.
7. Pg 14: In the first paragraph of the ADE section, clarify what 'within the episode' means. In this paragraph, it is also unclear how these sentences relate to ADE: 'Nasopharyngeal viral load increased in the first week and declined thereafter but clinical worsening was seen in many of the patients at week two, with virus shedding in stool and urine observed towards the end (26, 95). Many manifested with additional new lesions as original lesions improved. The timing of the appearance of new lesions correlated with IgG seroconversion, suggesting that pathology post week one was driven by the immune response rather than by uncontrolled viral replication.' It seems like timing of IgG is consistent with kinetics of protective response as described in earlier sections.
8. Pg 17: the differences between neut titers and binding titers might simply reflect differences in sensitivity of the assays
9. Great job with this review—very complete and informative.

Reviewer #3:

Remarks to the Author:

The SARS-CoV-2 pandemic has highlighted many gaps in our knowledge about the durability, specificity and functionality of antibodies in patients. We need to fill these gaps to accurately track the virus, to assess susceptibility to repeat infections and to guide vaccine research. As CoVs are a diverse family of respiratory viruses that have relatively recently been implicated as human pathogens, the immune response to these viruses is understudied. Nevertheless, over the years, a few groups have studied immunity to endemic human CoVs and the more pathogenic zoonotic viruses that have recently caused epidemics (SARS-1, MERS, SARS-2). In these early days of the SARS-2 pandemic, there is an urgent need to review and synthesize the existing literature on immunity to human CoVs to use as a foundation for understanding the role of antibodies in SARS-2.

This is a timely and comprehensive review of the existing literature on pathogenic human CoVs. The authors cover alpha and beta endemic human CoVs and the more pathogenic MERS and SARS groups of viruses including SARS-2. The manuscript is significant because of the careful and comprehensive analysis of the literature on previously recognized pathogenic CoVs. Currently, it is impossible to review the literature on SARS-2 because any attempt will be obsolete a month later.

The main strengths of the review are as follows:

1) The investigators have identified the most important questions and gaps related to antibodies and why this information is needed to contain the pandemic.

2) The authors have done a decent job of extracting data from studies done over a 50 year period with different viruses and research methods to compare and contrast the antibody response to these viruses. However, despite their best efforts, some of the topics remain a mystery because of contradictory conclusion and the difficulty of comparing poorly designed and controlled human studies.

3) Particular strengths of the review are the sections dealing with the potential for serological cross-reactivity between different CoVs, the impact of strain variation on protection and our current understanding of the prevalence of endemic human CoVs.

4) This review also has a supplementary tables that will be very helpful for other investigators looking for manuscripts on a specific topic.

I also have some suggestions for improving the manuscript.

1) I like the table at the beginning listing "Key Questions". However, the authors need to be more precise with the language used here. The authors use the terms immunity and protection loosely in these questions, whereas the review only deals with antibodies. For example, the first question is listed as "What are the kinetics of the immune response?" The review only deals with the "kinetics of the Ab response".

2) Supplementary table 1 list the different assays used to detect antibody. The description of neutralization assays is incomplete. Recombinant pseudotyped viruses with the Spike protein of SARS and MERS are being used by many groups to measure NAb under standard BSL2 conditions. I suggest adding this information and also considering if these platforms provide the same information as assays with WT virus.

3) Figure 2 in the main manuscript is incomprehensible because the lines for mean time of Ab appearance for the severe and mild cases cannot be seen against a background with too many colors.

4) Currently the literature on a role of antibodies in SARS pathogenesis is a mess with different human correlative studies reaching opposite conclusions and some laboratory studies that are not interpretable. In the discussion, the authors correctly point this out -"Multiple mechanisms for immunopathology have been suggested but no strong causal evidence exists. The extent to which antibody responses can in some circumstances contribute to disease severity is not known". I feel that figure 5 with potential mechanisms of ADE and the evidence for or against each mechanism is too speculative and not supported by the literature. The figure has the potential to contribute to more speculation and confusion rather than clarify.

Response to reviewers' comments

Reviewer #1 (Remarks to the Author):

The review discusses our current knowledge on Coronavirus antibody responses in the humans in relation to kinetics, cross reactivity and potential immunopathogenesis. The review is timely and important to add to our current understanding of immune responses to the current coronavirus SARS-CoV-2 which is needed for the pandemic management and release of lockdown measures, as well as counter measures and vaccine development; and thus, it is of interest to a wide community.

However, a main concern is: since the review focuses on the antibody responses to coronavirus, the term antibody has not been included in the search terms, which could have lead to missing some relevant research articles.

Response: We selected the search terms that we searched for in the original submission based on an initial screening of candidate terms. We found that “antibody coronavirus” and related terms returned a very high proportion of results that were not relevant and there was a large amount of overlap between the returned abstracts and ones returned from other searches included in our initial search.

That said, we do appreciate the critique that the reviewer has raised and want our review to be as comprehensive as possible. We have expanded our review to include “antibod* human coronavirus” and add to our review the additional resources that we identify from this screening. The results are restricted to publications up till March 20, 2020 to make the inclusion criteria uniform across search terms.

This large review has resulted in small changes throughout the review as we incorporated additional datasets and references. However, this additional review identified only a small number of sources that we had not identified and that were classified as meriting inclusion in the text.

* while the term antibody was missing from the search terms, the authors looked for "complement fixation", why was that specifically chosen? specially it is not the only antibody effector function.

Response: Complement fixation was a common serological measure of exposure and immunity to endemic coronaviruses in the first few decades of research on those viruses and thus we included it.

*Some Studies on antibody responses to SARS-CoV-2, kinetics of antibody isotypes, and cross reactivity were available online within the selected timeframe were not included in the review and could add up to the limited information available on SARS-CoV-2 immune responses at the time when the review was conducted, for example:

<https://doi.org/10.1101/2020.03.18.20038059>doi (PMID: 32267220)and
<https://doi.org/10.1101/2020.03.18.20038018> (PMID: 32350462) specially that the authors have referenced preprints (ref 35).

Response: Thank you for helping us look into what we may have missed. After checking these examples, we find that although the publications were made available prior but very close to our search time (March 20, 2020), the ePub dates were indeed later, hence why they did not show up in our search. In contrast, reference #35 (PMID: 32221519) was posted as a preprint on March 3, 2020. As the literature in SARS-CoV-2 is increasing at an unprecedented rate, we decided to adhere firmly to the results that show up on the date of database inquiry to ensure that we do not expand the scope of literature covered in a nonuniform way as it may introduce intrinsic biases and reduce the systematic aspect of our review.

* Where there any studies describing antibody responses against other structural proteins apart from the S and N proteins? This needs to be discussed in the manuscript and their role in cross-reactivity (specially that the authors mentioned the use of a truncated M protein in page 13 of the manuscript, with no previous mention to any responses against this protein)

Response: In restricting ourselves to results from the searches and direct follows of the literature, studies describing antibody responses against other structural proteins were limited. Three studies which touched upon this were found under the expanded search terms and have been added to the review.

* Would be useful to show the degree of homology of the four structural proteins among the 7 known coronaviruses in a table format. relevant for figure 4.

Response: The phylogeny in Figure 4 was based on the phylogeny resolved by Coronaviridae Study Group of the International Committee on Taxonomy of Viruses (PMID: 32123347) which concatenated multiple protein domains to derive an overall distance. The article was posted as a preprint on BioRxiv on February 11 and was a reference cited by another study that our search identified. We agree that the suggested table will likely be useful for the audience. However, in putting our best effort to keep our work a systematic review, we decided to refrain from performing substantial original analyses that go beyond summarizing results from other manuscripts.

Minor comments:

*Table 1 legend: refence to Figure 4 should be Figure 3

Response: Thank you for helping us catch this mistake. We have made the correction accordingly.

*Figure 2 legend: MERS-CoV (left row) correct to (upper row)

Response: Thank you for helping us catch this mistake. We have made the correction accordingly.

*Figure S5: upper panel is missing the x-axis numbering

Response: Thank you. We have made the correction accordingly.

*page 13, first paragraph, line5: IFA is used once and IF is used in the second time, please use one form for consistency and check this and other abbreviations throughout the manuscript.

Response: Thank you. We have made the correction and rechecked other abbreviations to ensure consistency.

*page 13, first paragraph, line5: which type of IFA showed the cross reactivity? is it a whole virus or a recombinant protein and if so which protein? please mention for clarification. because in the following sentence the authors jump to RBD similarity which is unlikely to be a reason for the observed cross reactivity.

Response: Thank you for this requested clarification. In the IFA assay described in this manuscript, Vero cells are infected with clinical isolates of SARS-CoV-1 or MERS-CoV. When 60-70% of cells have cytopathic effect, cells are harvested using trypsin, dried on slides, and fixed with acetone. Serum antibody was applied to the infected cells, and bound antibodies were detected with fluorescently-labeled anti-human IgG. Thus, the assay measures antibodies reactive to virions as well as to any viral protein present in the cells. We now specify that the assay measures virus to infected cells (change underlined):

“In the same study, animal handlers at a wildlife market in Guangzhou (n=94) with low-level prevalence of antibodies to SARS-CoV-1 (13.8% by IFA measuring antibody bound to infected cells, 4.3% by NT) had detectable antibodies toward MERS-CoV (2.2% by IFA) (K.-H. Chan et al. 2013).”

*page 13, first paragraph, line11: correct pandemic HCoV to endemic HCoV

Response: Thank you. We have made the correction accordingly.

Reviewer #2 (Remarks to the Author):

This is a fantastic review by Huang, Garcia-Carreras, Hitchings, Yang, Katzelnick and colleagues. I found the review to be very informative and well-written. The authors identified a number of critical papers that I was not aware of and I think that many people in the field will find this review useful. Some points to consider:

1. The review is focused on antibodies, but the authors should at least mention the involvement of cellular responses. For example, it is quite possible that T cell immunity is involved with cross-coronavirus protection. The authors should consider adding a small section that discusses immune responses other than antibodies.

Response: Thank you for your suggestion. We added text stating that our review was not focused on other immune responses including T cells and innate immune responses, although these may be very important to coronavirus infections, in the discussion/limitations. We did not include citations to example literature covering those responses even though we did come across some as our work would not ensure thorough evaluation of those topics and might end up pointing readers in random directions. The section discussing these responses was not added for the same reason. The added text is as below:

“Other aspects of immunity including the innate and cellular responses have been shown to be important in conferring protection at reexposure.”

Consider editing page 5 sentence ‘an exposure to a pathogens generates an immune response that changes over time and between individuals (antibody kinetics)’. Change ‘immune response’ to ‘antibody response’ since that what is examined in this review.

Response: Thank you. We have made the change as per suggested.

2. In the review, the authors correctly predicted that a lot of new SARS-CoV-2 data would become available before this is published. The authors might want to add a final paragraph that summarizes key finding published since their March literature searches, including papers describing the kinetics of IgG/IgM SARS-CoV-2 responses and development of diagnostics and therapeutic antibodies.

Response: Thank you for your suggestion. After multiple read throughs, we decided to remove information we obtained from a sample of publications published after our initial search date (March 20, 2020) for two main reasons. (1) Our search results are unlikely representative samples of the updated literature after March 20 and given the non-comprehensive assessment of the update. (2) By including them, we risk misleading the readers that our review did a comprehensive review or valid portrayal of the work that has been done on SARS-CoV-2. Given that other manuscripts have focused on the emerging literature on SARS-CoV-2 immune responses, we find that our review is better served by sticking with the results of our systematic search of the literature up to March 20, 2020. We do appreciate your suggestion in helping us identify ways to improve our work for the better of the community.

3. Edit Figure 2 X axis to be more precise ‘Time between symptom onset and detection’

Response: Thank you. We have made the change as per suggested.

4. Figure 3: what is each line and data point. Describe in text and legend.

Response: Thank you. Each point represents a measurement observation. The lines link observations of the same individual. We have added descriptions for the lines and points in the text and legend for clarity.

5. Page 7: HAI is not an antibody binding assay.

Response: We have rephrased this, including HAI as a separate category in our description of assays: “Hemagglutination inhibition assays (HAI), which measure the ability of antibodies in

sera to prevent binding of virus to red blood cells, have previously been used for coronaviruses but are no longer common.”

6. The authors should consider better describing the limitations of challenge studies. Very high doses of virus are used for these studies and could explain why there seems to be incomplete immunity in rechallenge studies.

Response: We have added a sentence stating that challenge studies may present an experimental exposure that is a different dose than natural exposures. “Of note, these experimental doses may differ from the amounts that people are exposed to in natural infections.”

7. Pg 14: In the first paragraph of the ADE section, clarify what ‘within the episode’ means. In this paragraph, it is also unclear how these sentences relate to ADE: ‘Nasopharyngeal viral load increased in the first week and declined thereafter but clinical worsening was seen in many of the patients at week two, with virus shedding in stool and urine observed towards the end (26, 95). Many manifested with additional new lesions as original lesions improved. The timing of the appearance of new lesions correlated with IgG seroconversion, suggesting that pathology post week one was driven by the immune response rather than by uncontrolled viral replication.’ It seems like timing of IgG is consistent with kinetics of protective response as described in earlier sections.

Response: We have revised the text from ‘within the episode’ to ‘within a single infection episode’ for clarity. In terms of the relatedness of the text that follows to ADE, we have revised the text to stress that new pathologies appeared at times when IgG began to rise and viral load declined which led the authors of the study to speculate that those pathologies may have resulted from presence of antibodies rather than of the virus.

8. Pg 17: the differences between neut titers and binding titers might simply reflect differences in sensitivity of the assays

Response: We have edited the sentence in question from “That the seroprevalence is markedly lower in that study could indicate lower prevalences of neutralizing antibody or lack of correlation between neutralizing and binding antibodies against endemic coronaviruses.” to “That the seroprevalence is markedly lower in that study could indicate lower prevalences of neutralizing antibody, lower sensitivity of neutralizing vs. binding assays, or lack of correlation between neutralizing and binding antibodies against endemic coronaviruses.”

9. Great job with this review—very complete and informative.

Response: Thank you for the encouragement. It is highly valuable to us.

Reviewer #3 (Remarks to the Author):

The SARS-CoV-2 pandemic has highlighted many gaps in our knowledge about the durability, specificity and functionality of antibodies in patients. We need to fill these gaps to accurately track the virus, to assess susceptibility to repeat infections and to guide vaccine research. As CoVs are a diverse family of respiratory viruses that have relatively recently been implicated as human pathogens, the immune response to these viruses is understudied. Nevertheless, over the years, a few groups have studied immunity to endemic human CoVs and the more pathogenic zoonotic viruses that have recently caused epidemics (SARS-1, MERS, SARS-2). In these early days of the SARS-2 pandemic, there is an urgent need to review and synthesize the existing literature on immunity to human CoVs to use as a foundation for understanding the role of antibodies in SARS-2.

This is a timely and comprehensive review of the existing literature on pathogenic human CoVs. The authors cover alpha and beta endemic human CoVs and the more pathogenic MERS and SARS groups of viruses including SARS-2. The manuscript is significant because of the careful and comprehensive analysis of the literature on previously recognized pathogenic CoVs. Currently, it is impossible to review the literature on SARS-2 because any attempt will be obsolete a month later.

The main strengths of the review are as follows:

1) The investigators have identified the most important questions and gaps related to antibodies and why this information is needed to contain the pandemic.

2) The authors have done a decent job of extracting data from studies done over a 50 year period with different viruses and research methods to compare and contrast the antibody response to these viruses. However, despite their best efforts, some of the topics remain a mystery because of contradictory conclusion and the difficulty of comparing poorly designed and controlled human studies.

3) Particular strengths of the review are the sections dealing with the potential for serological cross-reactivity between different CoVs, the impact of strain variation on protection and our current understanding of the prevalence of endemic human CoVs.

4) This review also has a supplementary tables that will be very helpful for other investigators looking for manuscripts on a specific topic.

I also have some suggestions for improving the manuscript.

1) I like the table at the beginning listing “Key Questions”. However, the authors need to be more precise with the language used here. The authors use the terms immunity and protection loosely in these questions, whereas the review only deals with antibodies. For example, the first question is listed as “What are the kinetics of the immune response?” The review only deals with the “kinetics of the Ab response”.

Response: Thank you for pointing this out. We have revised the questions to be specific to antibodies.

2) Supplementary table 1 list the different assays used to detect antibody. The description of neutralization assays is incomplete. Recombinant pseudotyped viruses with the Spike protein of SARS and MERS are being used by many groups to measure NAbS under standard BSL2 conditions. I suggest adding this information and also considering if these platforms provide the same information as assays with WT virus.

Response: Thank you for the suggestion. We have revised text under Limitations/Comments section to incorporate this information as “Gold standard and used often for confirmatory testing. Labor-intensive, expensive, requires BSL3. Can be done in BSL2 when using recombinant pseudotype virus-like particles. Agreement with wild type virus needs to be verified.”

3) Figure 2 in the main manuscript is incomprehensible because the lines for mean time of Ab appearance for the severe and mild cases cannot be seen against a background with too many colors.

Response: We agree. We have revised the figure such that the means are now plotted as points with interquartile intervals for better visibility.

4) Currently the literature on a role of antibodies in SARS pathogenesis is a mess with different human correlative studies reaching opposite conclusions and some laboratory studies that are not interpretable. In the discussion, the authors correctly point this out -“Multiple mechanisms for immunopathology have been suggested but no strong causal evidence exists. The extent to which antibody responses can in some circumstances contribute to disease severity is not known”. I feel that figure 5 with potential mechanisms of ADE and the evidence for or against each mechanism is too speculative and not supported by the literature. The figure has the potential to contribute to more speculation and confusion rather than clarify.

Response: We have conducted this review to provide a comprehensive foundation so that others can better piece together the vast efforts going into SARS-CoV-2 in various directions. We would regret very much if our work ends up misleading the community. That said, we think leaving out this summary will be rather detrimental as causal links are left implicitly drawn in the readers’ minds. To address the concern, we changed the line widths and color intensity of the

links to reflect the different uncertainties (experimental vs speculation). A legend has been added to further emphasize this point. We hope this will trigger the readers to realize the differential in weights of the evidence.